# A Strain of *Bacillus thuringiensis* Containing a Novel *cry7Aa2* Gene that Is Toxic to *Leptinotarsa decemlineata* (Say) (Coleoptera: Chrysomelidae)

**DOI:** 10.3390/insects10090259

**Published:** 2019-08-21

**Authors:** Mikel Domínguez-Arrizabalaga, Maite Villanueva, Ana Beatriz Fernandez, Primitivo Caballero

**Affiliations:** 1Institute for Multidisciplinary Research in Applied Biology-IMAB, Universidad Pública de Navarra, 31192 Mutilva, Navarra, Spain; 2Bioinsectis SL, Avda Pamplona 123, Mutilva, Navarra, Spain

**Keywords:** *Bacillus thuringiensis*, *cry* gene, toxins, Coleoptera, Colorado potato beetle, bioassay

## Abstract

The genome of the *Bacillus thuringiensis* BM311.1 strain was sequenced and assembled in 359 contigs containing a total of 6,390,221 bp. The plasmidic ORF of a putative *cry* gene from this strain was identified as a potential novel Cry protein of 1138 amino acid residues with a 98% identity compared to Cry7Aa1 and a predicted molecular mass of 129.4 kDa. The primary structure of Cry7Aa2, which had eight conserved blocks and the classical structure of three domains, differed in 28 amino acid residues from that of Cry7Aa1. The *cry7Aa2* gene was amplified by PCR and then expressed in the acrystalliferous strain BMB171. SDS-PAGE analysis confirmed the predicted molecular mass for the Cry7Aa2 protein and revealed that after in vitro trypsin incubation, the protein was degraded to a toxin of 62 kDa. However, when treated with digestive fluids from *Leptinotarsa decemlineata* larvae, one major proteinase-resistant fragment of slightly smaller size was produced. The spore and crystal mixture produced by the wild-type BM311.1 strain against *L. decemlineata* neonate larvae resulted in a LC_50_ value of 18.8 μg/mL, which was statistically similar to the estimated LC_50_ of 20.8 μg/mL for the recombinant BMB17-Cry7Aa2 strain. In addition, when this novel toxin was activated in vitro with commercial trypsin, the LC_50_ value was reduced 3.8-fold to LC_50_ = 4.9 μg/mL. The potential advantages of Cry7Aa2 protoxin compared to Cry7Aa1 protoxin when used in the control of insect pests are discussed.

## 1. Introduction

*Bacillus thuringiensis* (Berliner, 1915) (Bt) is an ubiquitous spore-forming bacterium which has been isolated from diverse habitats, including plant substrates, aquatic environments, animal excrements, poultry farms, dust from storage and mills, dead and living insects among other sources [1,2]. The entomopathogenic capacity of this organism lies in its ability to synthesize crystalline protein inclusions that have insecticidal activity when ingested by a susceptible host [3]. The toxicity spectrum of individual crystal proteins is usually limited to a number of species of the same order, although Bt as a species is toxic for an increasing number of insects belonging to different orders (Lepidoptera, Diptera, Coleoptera, Hymenoptera, Orthoptera, Hemiptera, etc.) as well as other invertebrates such as nematodes and mites [4,5]. Bt crystals are composed of Cry proteins, which are characteristically present in all Bt strains and show specific insecticidal activity, and Cyt proteins with cytolytic activity that are only found in the crystals of a small number of Bt strains. Cry proteins are the most diverse and numerous group of insecticidal Bt proteins. The Cry proteins described to date have been classified, according to the similarity of their respective amino acid sequences, into 78 families (from Cry1 to Cry78) and a larger number of subfamilies that comprise more than 800 toxins. An extensive number of these toxins have been described in recent years thanks to the availability of new molecular tools, as well as the growing interest in the discovery of Bt toxins with novel insecticidal characteristics [6].

The Cry proteins in the crystal are inactive protoxins, the toxicity of which depends on adequate solubilization and subsequent proteolytic digestion, as occurs in the midgut of a susceptible insect [7,8]. The peptides that are resistant to proteolytic digestion are active toxins that bind to specific receptors on the brush border membrane of the gut epithelium, generating pores that cause epithelial cell lysis, paralysis of the digestive system and finally insect death [9].

Cry proteins with demonstrated activity against coleopterans either belong to one of the 13 single-acting toxin families described so far (Cry1, Cry3, Cry6, Cry7, Cry8, Cry9, Cry10, Cry18, Cry22, Cry36, Cry43, Cry51, Cry55), or are part of the Cry23/Cry37 and Cry34/Cry35 binary protein families. The first Bt strain with specific toxicity to coleopterans was isolated in 1983 from *Tenebrio molitor* (Tenebrionidae) larvae [10]. The main component of the crystal of this strain and other related strains, are proteins of the Cry3 (Cry3A, Cry3B, and Cry3C) family with activity against several economically-important species, such as *Leptinotarsa decemlineata* (Say) (Coleoptera: Chrysomelidae) [11,12] and *Diabrotica virgifera* (Coleoptera: Chrysomelidae) [12,13].

The Cry7 family, while being an alternative to Cry3 proteins against certain coleopteran species of the genus *Cylas* [14], also represents a source of active toxins for other insect species from different orders for which few or no Bt toxins have been reported to date, such as the locust *Locusta migratoria manilensis* [15]. Cry7Aa1 was described as the first Cry7 protein with silent activity towards *L. decemlineata* larvae [16], meaning that the protoxin is not active by itself, but becomes active after solubilization and proteolytic digestion. Currently, the Cry7 family comprises a total of 37 toxins, which are grouped into several subfamilies (Cry7A–Cry7L), but insecticidal activity has only been reported for a few of them. For example, Cry7Ab3 was reported to be active against the spotted potato ladybeetle, *Henosepilachna vigintioctomaculata* (Coccinellidae) [17], whereas Cry7Aa1 showed high insecticidal activity in larvae of *Cylas brunneus* and *C. puncticollis* (Brentidae) [14]. The diversity of species susceptible to Cry7 suggests that this family of proteins may represent an interesting source of toxins with novel insecticidal properties.

The objective of this study was to determine the content of insecticidal genes present in the wild-type BM311.1 strain. The BM311.1 strain of Bt was isolated from an agricultural soil sample originating from a field in the Spanish province of Navarra as part of a country-wide screening program involving the isolation and characterization of Bt strains toxic to insects of agricultural importance [2]. This strain was selected because it was found to be toxic to coleopterans and contained at least one *cry7* gene in its genome (unpublished data). In the present study, the *cry7* gene present in this strain was identified and cloned, and its contribution to the insecticidal potency of BM311.1 was determined.

## 2. Materials and Methods

### 2.1. Bacterial Strains, Plasmids and Insect Culture Conditions

The acrystalliferous Bt strain BMB171^−^ was used as host strain for Bt protein expression [18]. For the routine gene cloning *Escherichia coli* XL1 blue was used, which was transformed with a slightly modified recombinant vector pSTAB [19] (pSTABr), engineered with the gene of interest. Both BM311.1 and BMB171-Cry7Aa2 strains were grown in CCY culture medium [20] under constant conditions of temperature (28 °C) and shaking (200 rpm). *E. coli* strains were cultured at 37 °C with shaking at 200 rpm in LB broth (1% tryptone, 0.5% yeast extract, and 1% NaCl, pH 7.0). When required for selective growth, the medium was supplemented with appropriated antibiotics at the following concentrations: erythromycin (Em), 20 mg/L, ampicillin (Amp), 100 mg/L.

A laboratory colony of *L. decemlineata* was established from adults collected from organic potato fields near Pamplona (Spain). This insect colony was maintained on potato plants in the insectary of the Universidad Publica de Navarra under controlled conditions of temperature, humidity and photoperiod (25 ± 1 °C, 70 ± 5% RH, and L16:D8 h) and was refreshed whenever it was possible to collect adults from the field.

### 2.2. Total DNA Extraction and Genomic Sequencing

Total genomic DNA (chromosomal + plasmid) was extracted following the protocol for DNA isolation from Gram-positive bacteria supplied in the Wizard^®^ Genomic DNA Purification Kit (Promega, Madison, WI, USA) and DNA library was prepared from total DNA and subsequently was sequenced by llumina NextSeq500 Sequencer (Genomics Research Hub Laboratory, School of Biosciences, Cardiff University, Cardiff, UK).

### 2.3. Identification of Potential Insecticidal Genes

Genomic raw sequence data were processed and assembled using CLC Genomics Workbench 10.1.1. Reads were trimmed, filtered by low quality and reads shorter than 50 bp were removed. Processed reads were *de novo* assembled using a stringent criterion of overlap of at least 95 bp of the read and 95% identity and reads were then mapped back to the contigs for assembly correction. Genes were predicted using GeneMark [21].

To assist the identification process of potential insecticidal toxin proteins, local BLASTP [22] was deployed against a database built in our laboratory including the amino acid sequences of known Bt toxins with insecticidal activity [23,24], as well as other proteins of interest such as the enhancins, metalloproteinases and mosquitocidal toxins available in GenBank.

The software PlasFlow was used for prediction of plasmid sequences from the assembled contigs [25]. Alignments of crystal protein sequences were performed using MUSCLE v3.8.31 [26]. Prediction of structural conserved domains was carried out using CD-search [27].

### 2.4. Amplification and Cloning of the cry7Aa2 Gene

A *cry7Aa*2 gene was amplified by PCR from Bt genomic DNA using primers Fw-*Nco*I (5′-TC*CCATGG*GTAATTTAAATAATTTAGGTGGATATGAAGATAGTAATAG-3′) and Rv-His-*Pst*I (5′-TC*CT GCAG*TTAATGATGATGATGATGATGACATAGCTCTTCCATCAAAAATAACTCTATAC-3′) and Phusion High-fidelity DNA polymerase (NEB, Ipswich, MA, USA). A 6xHis tag was placed at the N-terminal end of the gene. PCR products were purified by NucleoSpin^®^ Gel and PCR Clean Up kit (Macherey-Nagel Inc., Bethlehem, PA, USA) and ligated into the pJET plasmid (CloneJET PCR Cloning Kit, Thermo Scientific, Waltham, MA, USA). Ligation products were then electroporated into *E. coli* XL1 blue cells by using standard protocols [28]. Colony-PCR was applied in order to check positive clones from which plasmid DNA was purified, using the NucleoSpin^R^ plasmid kit (Macherey-Nagel Inc., Bethlehem, PA, USA), following the manufacturer’s instructions. Subsequently, pJET plasmids were verified by sequencing (StabVida, Caparica, Portugal) and digested with the appropriate combination of restriction enzymes to allow cloning into the pSTABr vector. Fragments from *Nco*I and *Pst*I were purified from agarose gels and ligated into a pre-digested pSTABr vector using the Rapid DNA ligation kit (Thermo Scientific) to obtain the recombinant plasmid pSTABr-*cry7Aa2.* Ligation products were then electroporated into *E. coli* XL1 blue cells by using standard protocols [28]. Positive clones were verified by colony-PCR and plasmids were purified and verified by digestion. Once pSTABr-*cry7Aa*2 was generated, it was introduced into the acrystalliferous Bt strain BMB171. pSTABr empty plasmid was also introduced into BMB171 strain as a negative control.

*Bacillus* electrocompetent cells were generated by modifying a previously described protocol [29]. Briefly, bacteria were grown in 300 mL of BHI broth at 28 °C under shaking conditions (200 rpm) until the culture reached an OD_600_ nm value of 0.4. Glycine was added to the culture at 2% and bacterial cells were incubated for another hour, at 28 °C, under shaking conditions (200 rpm). Bacterial cells were then kept on ice for 5 min, centrifuged for 10 min (9000 rpm, 4 °C) and the pellet was washed three times with F buffer (272 mM sucrose, 0.5 mM MgCl_2_, 0.5 mM K_2_HPO_4_, 0.5 mM KH_2_PO_4_, pH 7.2). The bacterial cells pellet was resuspended into 600 μL of ice-cold F buffer. Aliquots of 50 μL were stored at −80 °C. Plasmids were transformed into *Bacillus* by electroporation, as described previously [30]. Positive clones were selected by colony-PCR.

### 2.5. Production of Spores and Crystals from Wild and Recombinant Bt Strains

For both, the wild-type BM311.1 and the recombinant BMB171-Cry7Aa2, single colonies from LB plates were inoculated in 500 mL of CCY sporulation medium [20] supplemented with erythromycin for the recombinant strain and grown, at 28 °C, under shaking conditions (200 rpm). Crystal formation was observed daily under an optical microscope at the magnification of ×1000. After two or three days, when about 95% of the cells had lysed, the mixture of spores and crystals were collected by centrifugation at 9000× *g*, at 4 °C, for 10 min. After being washed with a saline solution (1 M NaCl, 10 mM EDTA), the mixture was resuspended in 10 mM KCl and kept at 4 °C until required. Protein quantification was performed by Bradford assay [31] using bovine serum albumin (BSA) as standard.

### 2.6. Analysis of Crystal Proteins

The composition of the crystals produced by the wild (BM311.1) and recombinant (BMB171-Cry7Aa2) strains were analyzed both in their natural form and once digested with midgut fluids from *L. decemlineata* or commercial trypsin. A group of 10 larvae of *L. decemlineata* fifth instar were forced to vomit to extract intestinal secretions and the pH of collected fluids was measured by MColorpHast^TM^ (Merck Millipore, Darmstadt, Germany). Aliquots of 25 μL of spore-crystal suspension were mixed with 5 μL of insect gut juice and incubated for 2 h at 37 °C and 200 rpm agitation. Another aliquot was solubilized in vitro in an alkaline solution (50 mM Na_2_CO_3_/10 mM DTT, pH 11.3) for 15 min at 37 °C and then digested with trypsin (Promega, Madison, WI, USA), using a 1/10 ratio (*w*/*w*) for 2 h at 37 °C. Samples of spores and crystals, both in their natural state and those previously digested by digestive fluids or trypsin, were mixed with 2x sample buffer (Bio-Rad, Hercules, CA, USA), boiled at 100 °C for 5 min, and then subjected to electrophoresis as previously described [32], using Criterion TGX™ 4–20% Precast Gel (BIO-RAD). Gels were stained with Coomassie brilliant blue R-250 (Bio-Rad) and then distained in 30% ethanol and 10% acetic acid. To identify the proteins expressed by the wild-type and recombinant strains, mass spectrometry analysis of the crystal-spore mixtures was carried out in the Proteomic Unity of the Navarrabiomed Centre (Pamplona, Spain), by clipping major bands of polyacrylamide gel (Figure 1). Protein identification was performed using Analyst 1.7.1 (Sciex, Concord, Ontario, Canada) and spectra files were processed through Protein Pilot Software (v.5.0-Sciex) using ParagonTM algorithm (v.4.0.0.0) for database search, ProgroupTM for data grouping, and searched against the concatenated target-decoy UniProt *Bacillus thuringiensis* database.

### 2.7. Leptinotarsa Decemlineata Bioassays

The insecticidal activity of the spore and crystal mixtures of both Bt strains, BM311.1 and BMB171-Cry7Aa2, as well as the BMB171-Cry7Aa2 crystal proteins previously solubilized and trypsinized in vitro, were tested against *L. decemlineata*. The concentration-mortality responses were subsequently determined using five different protein concentrations, ranging from to 0.24 to 150 µg/mL, in order to estimate the 50% lethal concentration (LC_50_). In all cases, small disks of potato leaves were dipped in the spore/crystal mixture, allowed to air dry and individually placed in wells of a tissue culture plate containing a layer of 1.5% (*w*/*v*) agar to prevent desiccation. Control leaf disks were treated identically but were not inoculated with crystal proteins. A 6–12 h old larva of *L. decemlineata* was placed in each well and incubated at 25 ± 1 °C. Insect mortality was recorded 4 days later. For each protein concentration and the control, groups of 24 larvae were treated and the complete bioassay was performed on three occasions using different batches of insects from the colony. The results were subjected to Probit analysis [33] using the POLO-PC program [34].

### 2.8. Nucleotide Sequence Accession Number

The nucleotide sequence data reported in this paper have been deposited in the GenBank database under accession number SSWY00000000 for the BM311.1 genome and MK840959 for the *cry7Aa2* gene.

## 3. Results

### 3.1. Draft Genome Sequence of the Bacillus thuringiensis BM311.1 Strain

The reads obtained from the genomic DNA of the BM311.1 strain were assembled and produced 359 contigs containing a total of 6,390,221 bp, with a maximum scaffold size of 276,646 bp, a N50 length of 55,361 bp, and 33.6% GC content. The genome of strain BM311.1 contains three *cry*-like ORFs, two of them located in different chromosomal contigs while the third one was present on a plasmid. The two chromosomal ORFs shared less than 30% identity between them and, for each of them, the closest Cry protein, with less than 20% identity, was the product of the *cry60Aa* gene, previously described by Sun et al. [35] (Table 1). Neither of these two proteins could be classified in any of the Cry families currently described in the Bt full toxin list [23]. The third ORF identified shared 98% identity with the *cry7Aa1* gene [16]. Therefore, this new *cry* gene was classified within the c*ry7* family and according to the current nomenclature, has been named Cry7Aa2. In addition, other potential virulence factors: one mosquitocidal-like toxin, two bacillolysins and four peptidase M4 genes were detected in the genome of BM311.1 (Table 1).

### 3.2. Characterization of Cry7Aa2

The recombinant BMB171-Cry7Aa2 strain harboring the *cry7Aa*2 gene was able to form large inclusion bodies that could be observed under the optical microscope. The predicted molecular mass of the Cry7Aa2 protein was 129.4 kDa, which corresponds to the bands of approximately 130 kDa generated by SDS-PAGE for the spore/crystal proteins produced by both BM311.1 and BMB171-Cry7Aa2 (Figure 1). Proteomic analysis of the polyacrylamide gel bands of both strains was carried out and Cry7Aa1 protein was identified in both cases. When the Cry7Aa2 crystal protoxin was solubilized and activated in vitro with commercial trypsin, a fragment of approximately 62 kDa was produced. However, when treated with acidic digestive fluids (pH 5–6) of *L. decemlineata* larvae, a major band of approximately 60 kDa was detected.

The alignment of the deduced amino acid sequence of Cry7Aa2 with the known Cry7Aa1 protein revealed that the new protein had 28 different amino acids, which appeared randomly distributed throughout the amino acid sequence of the protoxin (Figure 2). The analysis of the primary structure of Cry7Aa2 revealed the presence of eight conserved blocks and the classical structure of three domains (Figure 2). Two of eleven changes within domain I were located in the second block of conserved amino acids and three changes were located in non-conserved blocks within domain II. Of the six changes detected within domain III, one of them was located in the fourth conserved block. Finally, only two different residues were located in C-terminal amino acid sequence, out of the three domains of the Cry7Aa2 protein.

### 3.3. Insecticidal Activity of Cry7Aa2 for L. decemlineata

To determine the insecticidal activity of Cry7Aa2, mixtures of spores/crystals from the wild type and recombinant strains, and solubilized and trypsinized proteins from the trypsin activated (TA) strain were used. The protein concentrations produced by all the strains were normalized and an equal amount of each of them was used to run toxicity tests on newly hatched *L. decemlineata* larvae. A recombinant acrystalliferous strain carrying an “empty” plasmid that was unable to produce crystals, was introduced as a negative control. Following ingestion of crystal and spore mixtures from both the wild-type BM311.1 and the recombinant BMB171-Cry7Aa2 strains, *L. decemlineata* larvae showed high levels of mortality, whereas none of the control insects died. The estimated LC_50_ values for BM311.1 and BMB171-Cry7Aa2 strains were 18.8 and 20.8 μg/mL, respectively (Table 2). However, the LC_50_ value for Cry7Aa2 protoxins activated with commercial trypsin was 3.8-fold lower than when ingested as a component of the crystal produced by the recombinant BMB171-Cry7Aa2 (Table 2).

## 4. Discussion

The complete genome of *B. thuringiensis* wild strain BM311.1 was sequenced and annotated. This Bt strain contains three putative insecticidal Cry proteins. One of them showed 98% amino acid identity with Cry7Aa1 [16] and was classified accordingly as Cry7Aa2 [23]. This new gene was cloned and sequenced and the natural protoxin for which its codes was found to have insecticidal properties against larvae of *L. decemlineata*. The other two ORFs shared less than 30% identity and less than 20% identity with Cry60Aa, the protein with which they showed the highest identity. These genes need to be cloned and the corresponding proteins characterized in detail to determine their insecticidal potential.

SDS-PAGE analysis showed that the main crystal composition was represented by a predominant protein band of a molecular mass of approximately 130 kDa. According to the *cry* gene content of BM311.1 this band could only correspond to the expression of the *cry7Aa2* gene with a predicted molecular mass of 130 kDa. However, this analysis did not detect the putative proteins encoded (predicted molecular weight of 33–35 kDa; Table 1) by the other two ORFs. Often the proteins encoded by certain *cry* genes are not part of the proteins that make up the crystal of a given Bt strain. This may be because these genes are not expressed, or the level of expression is below the detection capacity of the SDS-PAGE technique. Another possible explanation is that the proteins are expressed and secreted into the medium in which the bacterium grows, as is the case with Cry1I proteins [36].

The complete ORF of the new *cry7Aa2* gene was cloned and expressed in the acrystalliferous strain BMB171^−^. SDS-PAGE analysis of the spore/crystal mixture of this recombinant strain generated a predominant 130 kDa protein band, similar to that generated by other proteins of the Cry7 family [16,17]. When *cry7Aa2* was expressed in BMB171, the resulting protein formed a parasporal crystal of a larger size than the one produced by the wild-type BM311.1 strain. This result suggests that BMB171 strain may contain chaperones that improve the expression of the Cry7Aa2 protein, as has already been reported for other Cry proteins [37,38]. It could also be attributed to the presence of transcriptional or posttranscriptional regulation in the wild type BM311.1 strain [39]. It should be noted that SDS-PAGE shows a slightly smaller band in size for the Cry7Aa2 recombinant protein when compared to its wild-type counterpart that can be attributed to different protein folding (Figure 1, Lane 2). However, proteomic analysis confirmed that both strains expressed the same Cry7Aa2 protein. The natural Cry7Aa*2* and recombinant Cry7Aa*2* showed comparable toxicity levels towards *L. decemlineata* larvae in addition to a similar proteolytic processing with the insect’s digestive fluids or commercial trypsin (data not shown).

Amino acid sequence analysis of Cry7Aa2 showed similar characteristics to those of other proteins in the Cry7 family and differed in only 28 amino acid residues from the reference protein, Cry7Aa1 [16]. The difference in a few amino acids can produce very important changes in the insecticidal properties of Cry toxins. For example, a single amino acid variation between the Cry1Ia1 and Cry1Ia2 toxins has been associated with the different host spectra of these two proteins [40].

Insect bioassays revealed that the Cry7Aa2 protoxin was active against newly hatched larvae of *L. decemlineata* when inoculated as a crystal component produced by both the wild-type BM311.1 and the recombinant BMB171-Cry7Aa2. In contrast, Lambert et al. [16] reported that the Cry7Aa1 protein showed silent activity towards larvae of *L. decemlineata*, i.e., the natural Cry7Aa1 protein was not toxic when ingested as part of the crystal, but was toxic once solubilized and activated in vitro. These results suggest that the lack of toxicity can be likely attributed to the lack of solubilization in the acidic digestive fluids of the coleopteran gut. In contrast, others have reported toxicity of Cry7Aa1 protoxin [14] and Cry7Ab3 protoxin [41] in other species of Coleoptera that also had acidic digestive fluids, so it seems reasonable to assume that the solubilization of Cry7 proteins must have involved factors other than pH. Although the solubilization of Cry proteins and their subsequent proteolytic digestion are determinants of toxicity, the interaction between the toxin and the appropriate midgut receptors is also necessary for the formation of the lytic pore [42]. Following incubation with insect digestive juices the Cry7Aa2 protoxin produced a single fragment of approximately 60 kDa. In contrast, when this protoxin was digested with trypsin it produced a single fragment of approximately 62 kDa. Interestingly, the fragment showed a 3.8-fold higher toxicity than when the toxin was activated in the midgut of *L. decemlineata* larvae. This indicated that the fragment was derived from Cry7Aa2 and that the different activation method may be the reason behind the augmented potency.

The fact that the natural protein Cry7Aa2 was toxic when it formed part of the crystal of the BM311.1 strain represents a clear advantage over the Cry7Aa1 protoxin for its use in bioinsecticide applications. However, both the Cry7Aa1 and Cry7Aa2 proteins can be efficiently exploited in the construction of transgenic plants since this technology allows the peptide fragment encoding the toxin to be expressed directly instead of using the sequence coding for the protoxin. The Cry3 family of proteins has been frequently used as the active ingredient of a bioinsecticides as well as for the construction of transgenic plants for the control of coleopteran pests [12,43,44]. Although resistance to Bt has not yet reached the prevalence reported in insects exposed to chemical pesticides, there is evidence that the extended use of Bt toxins may accelerate the appearance of insect resistance [45,46], so that the characterization of novel Bt toxins is an issue that will likely continue to attract the attention of insect pathologists and pest control researchers for the foreseeable future.

## 5. Conclusions

A new *cry7Aa2* gene that codes for a toxic protein for larvae of *L. decemlineata* has been identified. The most important finding is that Cry7Aa2, in addition to expanding the range of Bt toxins available for the control of coleopterous pests, can be used directly in its crystallized form while Cry7Aa1 needs to be previously solubilized *in vitro*. Future studies will focus on determining the host spectrum of Cry7Aa2 and Cry7Aa1 and elucidate the molecular basis for the different solubilization of these proteins.

## Figures and Tables

**Figure 1 insects-10-00259-f001:**
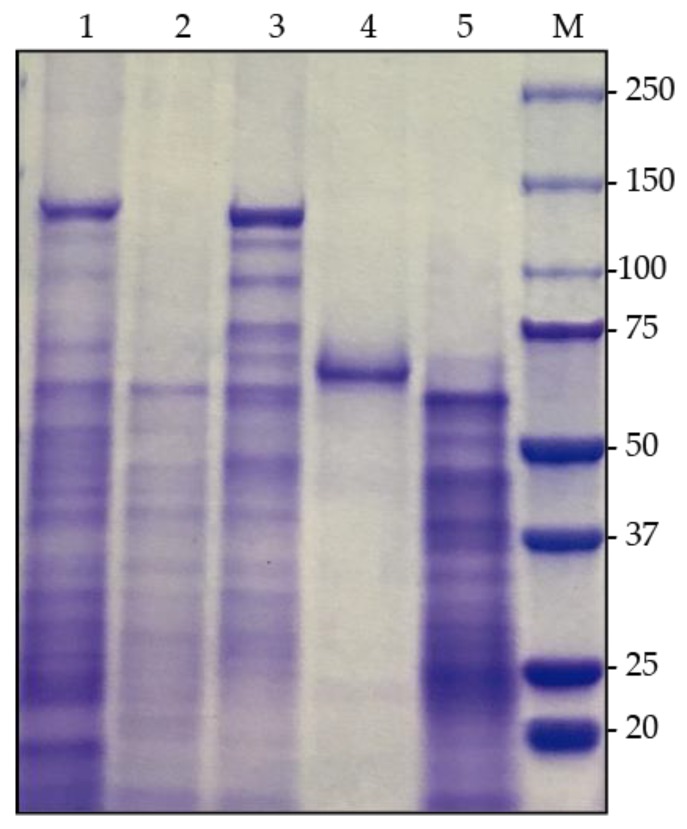
SDS-PAGE analysis of spore and crystal proteins from Bt strains. (**1**) Bt strain BM311.1; (**2**) recombinant Bt strain BMB171-pSTABr; (**3**) recombinant Bt strain BMB171-Cry7Aa2; (**4**); BMB171-Cry7Aa2 crystal protein solubilized and digested with trypsin; (**5**) BMB171-Cry7Aa2 crystal protein digested with digestive fluids from *L. decemlineata*; (**M**) molecular weight marker in kDa.

**Figure 2 insects-10-00259-f002:**
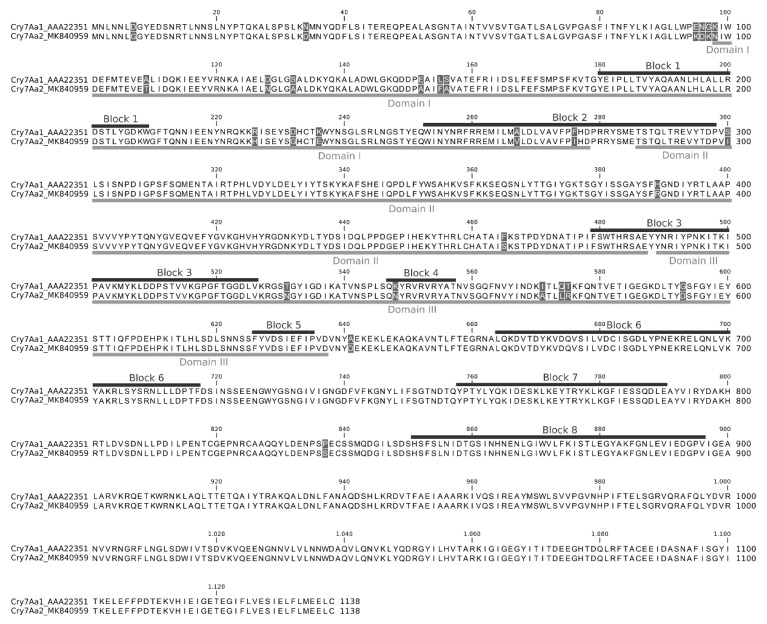
Alignment of the deduced amino acid sequence of Cry7Aa1 and Cry7Aa2. Non-conserved amino acid residues are shaded. Conserved blocks and structural domains are indicated in dark and light gray horizontal bars, respectively.

**Table 1 insects-10-00259-t001:** Insecticidal protein content of *Bacillus thuringiensis* BM311.1.

Target Database	Identity (%)	MW (KDa)	Length (Nº Residues)	Predicted Location
Cry7Aa1	98	129	1138	Plasmid
Cry60Aa1	18	35	322	Chromosome
Cry60Aa3	19	33	303	Chromosome
Mtx-like	94	57	515	Plasmid
Bacillolysin	99	61	556	Chromosome
Bacillolysin	96	98	893	Plasmid
Peptidase M4	99	65	583	Chromosome
Peptidase M4	99	62	567	Unclassified
Peptidase M4	99	62	552	Plasmid
Peptidase M4	99	61	566	Chromosome

**Table 2 insects-10-00259-t002:** Insecticidal activity of Bt strains. LC_50_ values and relative potency of Cry7Aa2 protoxin when ingested, by newly hatched larva of *L. decemlineata*, as a component of crystals produced by BM311.1 or BMB171-Cry7Aa2 or after toxin activation with trypsin.

Bt Strains/Protein	Regression Lines		LC_50_ (µg/mL)	Goodness of Fit		Relative Potency ^(b)^	Fiducial Limits (95%)	
	Slope ± SE	Intercept ± SE		χ^2^	df ^(a)^		Lower	Upper
BM311.1	0.63 ± 0.10	4.19 ± 0.13	18.89	0.99	3	1		
BMB171-Cry7Aa2	1.16 ± 0.19	3.46 ± 0.29	20.80	1.18	3	0.91	0.39	2.13
BMB171-Cry7Aa2-TA ^(c)^	1.99 ± 0.54	3.61 ± 0.53	4.93	1.02	2	3.83	1.57	9.33

^(a)^ df; ^(b)^ The relative potency was expressed as the ratio of the LC_50_ value for each treatment and the LC_50_ value of wild-type BM311.1; ^(c)^ TA: Trypsin Activated.

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
