# Peer review of "A Strain of Bacillus thuringiensis Containing a Novel cry7Aa2 Gene that Is Toxic to Leptinotarsa decemlineata (Say) (Coleoptera: Chrysomelidae)"

_insects, 2019, doi:10.3390/insects10090259_

Round 1

Reviewer 1 Report

Please find comments in PDF file.

Author Response

25 July 2019

Dear Reviewer

We have revised the MS (insects-503702) in line with the comments of the four referees. The changes in the revised MS are outlined below. Line numbers in the Author response section refer to the revised manuscript. The Bacillus thuringiensis nomenclature committee has recently assigned our protein the name Cry7Aa2. So, we have changed the name of the protein from Cry7Aa-like to Cry7Aa2 in the whole document, including figures and tables.

Referee comments

Author response

Reviewer 1

Line 16-17. Is it Cry7Aa2 or Cry7Aa1?

The protein has been assigned the name Cry7Aa2 by   the Bt nomenclature committee. It differs from Cry7Aa1 (lines 16-17)

Line 38. B. thuringiensis

Done.

Line 39. (Lepidoptera, Diptera,   Coleoptera, Hymenoptera, Orthoptera, Hemiptera, etc.

Revised in the manuscript: (Lepidoptera, Diptera, Coleoptera,   Hymenoptera, Orthoptera, Hemiptera, etc.) (line 39)

Line 57. ref

Reference added in the manuscript

Line 204. B. thuringiensis

Changed in the manuscript to Bt

We thank all the reviewers for their careful and constructive revision of the manuscript.

We hope that the MS is now in an acceptable form for publication in Insects.

Thank you for your help with this paper. Please let me know if you feel that additional changes are required.

Best regards,

Primitivo Caballero

Reviewer 2 Report

This new version is fine except that the authors should define silent activity at first use, rather than in the discussion as suggested in my previous review.

Author Response

25 July 2019

Dear Reviewer

We have revised the MS (insects-503702) in line with the comments of the four referees. The changes in the revised MS are outlined below. Line numbers in the Author response section refer to the revised manuscript. The Bacillus thuringiensis nomenclature committee has recently assigned our protein the name Cry7Aa2. So, we have changed the name of the protein from Cry7Aa-like to Cry7Aa2 in the whole document, including figures and tables.

Referee comments

Author response

Reviewer 2

This new version is fine except that the authors should  define silent activity at first use, rather than in the   discussion as suggested in my previous review.

Silent activity has been defined in lines 67-68

We thank all the reviewers for their careful and constructive revision of the manuscript.

We hope that the MS is now in an acceptable form for publication in Insects.

Thank you for your help with this paper. Please let me know if you feel that additional changes are required.

Best regards,

Primitivo Caballero

Reviewer 3 Report

The explanations for the data (Fig. 1 and Table 2) that (I said) does not support some of the authors conclusions are unconvincing. Addition of a His-tag does not increase protein mobility in SDS-PAGE (that I have ever seen), it only decreases it! This is because SDS binds to hydrophobic regions of proteins, and a His-tag is not hydrophobic. Thus, it is unlikely that the major ~135 kDa protein in lane 1 is Cry7Aa2.

The authors' explanation for why the bioassay was not biased due to unequal crystal protein  (presumably Cry7Aa2) production doesn't make any sense at all. If equal amounts of protein from each strain (crystals + spores) were used in the bioassay then it had to have differing amounts of cry7Aa2, if Figure 1 is to be believed. Incidentally, it should be stated in the legend to Fig. 1 how the gel was loaded, for example, equal vol, equal protein, or equal ...?

An important new piece of data/information is in the revised manuscript, but it is in a statement in the text (lines 236-239) rather than in the figures, and it is that the BMB171 strain was also mock-transformed with an empty plasmid and it was not toxic. That is a very important strain, and it should be included in Fig. 1 to help prove the ~130 kd protein in lane 2 is in fact Cry7Aa2. Moreover, bioassay results that include the mock-transformed BMB171 strain should be presented, and before the table of LC50 values are presented (Table 2). That way the authors can at least claim that the protoxin form of Cry7Aa2 is in fact toxic. However, I also recommend deleting all the data with the BM311.1 strain; it is not necessary and it is not definitive. Part of the problem with it is that the authors do not know how much Cry7Aa2 is produced by this strain, if any! Indeed, the comparison of LC50 values between BM311.1 and BMB171-CryAa2 is nothing but perplexing; the latter should be much more toxic based on Fig. 1 and the described methods, yet it isn't. 

That is the best advice I can give you.             

Reviewer 4 Report

The paper shows an interesting work of detectection and characterization of a new Bt protein. The protein is toxic againts Leptinotarsa decemlineata, a species of difficult control. The paper is well written and ordered, but some points need revision:

Main considerations:

1) In the title, the authors say that the new protein is “highly toxic” against Leptinotarsa decemlineata, but after, there is not information for CL50 comparison. The bibliography (Park, 2009) says that CL50 of Cry3Aa and Cry3B are 3.86 and 6.86 micrograms/ml, respectively. In your paper, only Cry7Aa2-TA has a similar CL50. So, may be in the title you have to change “higly toxic” by “toxic”, or provide some date showing the high toxicity of the protein for L. decemlineata.

2) How many times were performed protein activation experiments? Were the results consistent and repetitive?

3) Page 8. Line 277 to 283. When Cry7Aa2 was expressed in plasmid BMB171. The resulting protein is not bigger than the wild type strain, as you can see in the gel, that is not normalized and is difficult to compare. In any case,  the band of the recombinant strain is located  under the wild-type. Moreover, chaperones have usually 20 kDa, so the difference in weight should be larger. Authors have to explain this point better.

Minor considerations:

4) Page 3. Line119. Because the cry7Aa2 is the first one and a specific gene, change “A cry7Aa2 gene”…by “The cry7Aa2 gene…

5)  Page 4. Line169. Eliminate “rearing”

6)  In table 2: What means “df”? Meaning is not indicated.

7)  Page 8. Line 274. I think than when you say (Figure 1, Line 3), you must to say (Figure 1, Lane2). Because Lane 3 is the recombinant processed not the native recombinant.

8) Page 10. Line 416. Change d-endotoxin by d-endotoxin

In supplementary material:

9)  Page 15 and page 16 TestCry7Aa2 (David)…eliminate “David”.

10) In the entire document the label “MB39_NA311” appears. In the Introduction authors say that BM311.1 was collected in Navarra (Spain). As de numbers are same, we can suppose that NA311 is the original name of the strain. It should be better write the same name in all the documents.

Round 2

Reviewer 3 Report

The authors have provided a much improved Figure 1 and verified the identity of the protein in the gels. It is now a solid paper in my opinion, and I recommend publication.

This manuscript is a resubmission of an earlier submission. The following is a list of the peer review reports and author responses from that submission.

Round 1

Reviewer 1 Report

Some of the the authors' most important conclusions are not fully supported by the data, and therefore alternative explanations have not been excluded. The concern surrounds Figure 1 and Table 2. In Figure 1, there is no protein in the wild-type strain (lane 1) that is the same size of the one expressed in the recombinant strain (lane 2); there is one a little larger in the wild-type but it is not the same size. Hence, where is the protein from the crystal the authors' observed in the wild-type strain? The authors also repeatedly refer to crystals with single bands or single major bands, but they don't purify any crystals or inclusions.

Regarding Table 2, what was actually measured for the LC50 determination? Why does the recombinant strain, which vastly overproduces the protein because of the pSTAB expression plasmid, have the same toxicity as the wild-type strain? Maybe the cry7A-like protoxin is not toxic at all, and the wild-type strain is not even expressing the cry7Aa-like gene, and the toxicity really comes from some other toxin altogether? Purifying the crystals and assaying them should go a long way toward answering these questions.

The trypsin-treated protein (Fig. 1, lane 3) does seem to be pure, trimmed, recombinant cry7Aa-like protein, but that should be verified in some way. So, the good news is that protein, at least, shows some toxicity toward the larvae (yeah!). Maybe that exact protein, once its ends are determined,  could be expressed in some system to make a biological toxin against these larvae?

The Discussion section is also very confusing (paragraphs 2, 3, 5 and 6) but I think that is largely because the data does not support the conclusions as noted above.

Reviewer 2 Report

Novel Bt toxins with new activity, even if this is a subtle change in host range, are of broad interest and can understand mode of action in this important group- as well as providing valuable reference for those interested in biocontrol.

This paper has suitable methods for the descrption of a new toxin and I recommend acceptance.

Would be nice to see activity of cloned Cry7Aa in order to compare activity  as insect population can affect bioassay results quite dramatically in some species.

My other comment is that the introduction & methods in particular need careful proofing for grammar.   I have found quite a a few typos and small errors listed below.

Minor comments

Line 34-35  “one or more crystalline proteins”  

Line 38-  I do wonder about the quality of the data on some of these orders.  Lep Dip & Coleoptera are not in doubt but if you are claiming activity for some of the novel groups I think you should supply references.

Line 49, comma after protoxins

Line 52 -3 ” the gut epithelium” not gut epithelial  ; cell lysis not cells lysis

Insect death not “the insect death”

Line 56 “or well on the”  not sure what this means

Line 57 “Coleoptera” or coleopterans not coleopteran.

Line 66 what is “silent activity” ????  this is only defined on line 280

Line 90 – check journal style for formatting of units – mg/l is probably correct

Reviewer 3 Report

Did the author used overlapping PCR for amplification of the gene?

One stretch of amplification may show some error though it was high fidelity. 

Abstract line no. 22-23 about LC50.

Did the recombinant product compare with any commercial product?